

# Permutation Entropy and Complexity Analysis of Large-scale Solar Wind Structures and Streams

Emilia Kilpua[1], Simon Good[1], Matti Ala-Lahti[2,1], Adnane Osmane[1], and Venla Koikkalainen[1]

[1]Department of Physics, University of Helsinki, P.O. Box 64, FI-00014 Helsinki, Finland
[2]Department of Climate and Space Sciences and Engineering, University of Michigan, 2455 Hayward St., Ann Arbor, MI 48109-2143, USA

**Correspondence:** Emilia Kilpua (emilia.kilpua@helsinki.fi)

**Abstract.** In this work, we perform a statistical study of magnetic field fluctuations in the solar wind at 1 au using permutation entropy and complexity analysis. Slow and fast wind, magnetic clouds, interplanetary coronal mass ejection (ICME)-driven sheath regions and slow–fast stream interaction regions (SIRs) have been investigated separately. Our key finding is that there are significant differences in permutation entropy and complexity values between the solar wind types at larger timescales

and little difference at small timescales. Differences become more distinct with increasing timescale, suggesting that smaller-scale turbulent features are more universal. At larger timescales, the analysis method can be used to identify localized spatial structures. We found that fluctuation properties in compressive structures (sheaths and SIRs) exhibit a clear locality. Our results shows that, in all cases apart from magnetic clouds at largest scales, solar wind fluctuations are stochastic with the fast wind having the highest entropies and low complexities. Magnetic clouds in turn exhibit the lowest entropy and highest complexity,

consistent with them being coherent structures in which the magnetic field components vary in an ordered manner. SIRs, slow wind and ICME sheaths are intermediate to magnetic clouds and fast wind, reflecting the increasingly ordered structure. Our results also indicate that permutation entropy – complexity analysis is a useful tool for characterizing the solar wind and investigating the nature of its fluctuations.

## 1 Introduction

The study of multi-scale magnetic field fluctuations is an active research area in space, astrophysical and laboratory plasmas. One of the few natural environments in which it is possible to study them with direct measurements is the collisionless solar wind that incessantly streams from the Sun and fills the heliosphere (e.g., Bruno and Carbone, 2013). Solar wind fluctuations are generally thought to arise from waves, turbulence and coherent structures, but many open questions regarding their nature and evolution prevail. So-called 'mesoscale' solar wind structures, corresponding to structures with spatial extents of approximately

5-10,000 Mm and temporal scales ranging from $\sim$ 10 s to 7 h near Earth's orbit ($\sim$ 1 AU), have also recently been brought to the centre of attention due to their importance in solar wind formation and evolution, and their space weather impacts (e.g., Viall et al., 2021).

Outward-propagating incompressible Alfvénic fluctuations from the Sun are a common feature of fast solar wind streams (e.g., Belcher and Davis, 1971). Their non-linear interaction with inward-directed Alfvén waves generated locally in interplan-



etary space (e.g. Chen et al., 2020) are believed to drive a turbulent cascade of energy from large to small scales, where energy
finally dissipates and heats the solar wind (Smith and Vasquez, 2021). The slow solar wind is also turbulent but with a more
variable structure and a higher occurrence of coherent intermittent structures (e.g., Bruno et al., 2003; Wawrzaszek and Echim,
2021). Knowledge of the properties and nature of magnetic field fluctuations is also important for understanding large-scale
heliospheric structures, such as interplanetary coronal mass ejections (ICMEs) and their sheaths (Kilpua et al., 2017a), and

slow–fast stream interaction regions (SIR; e.g., Richardson, 2018), as well as for understanding how energy is transferred
through their boundaries. In addition, magnetic fluctuations have an important role in the acceleration and transport of solar
energetic particles (SEPs; e.g., Oughton and Engelbrecht, 2021), and highly fluctuating solar wind is also considered more
geoeffective (e.g., Borovsky and Funsten, 2003; Osmane et al., 2015; Kilpua et al., 2017b; Telloni et al., 2021; Han et al.,
2023; Dai et al., 2023).

Permutation entropy analysis (Bandt and Pompe, 2002) and Jensen-Shannon complexity analysis (Rosso et al., 2007) are
powerful tools for investigating fluctuations. They have been used in widely-ranging contexts and also recently in a few space
plasma physics studies (Weck et al., 2015; Weygand and Kivelson, 2019; Osmane et al., 2019; Good et al., 2020; Olivier et al.,
2019; Kilpua et al., 2022). The determination of permutation entropy and Jensen-Shannon complexity is based on investigating
the occurrence of permutation (or ordinal) patterns in time series. An ordinal pattern is formed by a set of subsequent values

in a time series separated by time lag $\tau$ and it thus gives information on the relation between the values forming the pattern.
Varying $\tau$ allows fluctuations over multiple time scales to be investigated. The number of elements in an ordinal pattern is
called the embedded dimension, $d$, and the factorial of $d$ gives the number of possible permutations. The frequency at which
different ordinal patterns occur in a time series determines its entropy and predictability. For example, if only a few ordinal
patterns are present (i.e., all other permutations have zero probability), permutation entropy is close to zero, signifying high

predictability or knowledge of the underlying process. A situation in which all ordinal patterns occur with the same probability
yields the maximum entropy state, signifying low predictability.

However, permutation entropy cannot yield information about the randomness of the patterns or the structural complexity of
time series. Complexity is related to how far the distribution formed by all permutations in the time series is from the maximum
entropy (uniform) distribution (e.g., Zanin and Olivares, 2021). Both highly ordered cases (e.g., periodic fluctuations like sine

waves) and random cases (e.g., rolling of a dice, white and pink noise) have low complexities. Note that the former case has
lower entropy and the latter close to the maximum entropy. In between the zero and maximum entropy cases, complexity
can have a range of values. Maximal complexities are associated with chaotic fluctuations that are structured but have lower
predictability.

Those few studies made in the solar wind using complexity and entropy analysis have indicated that magnetic field fluctua-

tions are stochastic (Weck et al., 2015; Weygand and Kivelson, 2019; Good et al., 2020; Kilpua et al., 2022). Weck et al. (2015)
analyzed both laboratory plasmas and solar wind. Their study suggests that solar wind fluctuations represent fully developed
turbulence, while fluctuations in the investigated laboratory settings were weakly turbulent or not even truly turbulent. Wey-
gand and Kivelson (2019) investigated the complexity of solar wind magnetic fluctuations in ICMEs and SIRs using both Wind
and Ulysses data at distances from 1 to 5.4 au. In all cases fluctuations were stochastic, and they found that their complexity



decreased and entropy increased with increasing heliospheric distance from the Sun. Good et al. (2020) and (Kilpua et al., 2022) were both case studies of a slow ICME-driven sheath.

In this paper we investigate the occurrence of ordinal patterns, entropy and complexity in different types of solar wind. The categories include slow and fast solar wind, ICME sheaths, SIRs and magnetic clouds. In addition, we investigate the memory of the time series extracted from these structures by analysing their magnetic field fluctuation scaling properties with the Hurst exponent.

## 2 Methods and approaches

### 2.1 Data and event selections

We here use 3 s magnetic field data from the Magnetic Fields Investigation (MFI; Lepping et al., 1995) fluxgate magnetometer on board the *Wind* spacecraft (Ogilvie et al., 1995). The data is analyzed in Geocentric Solar Ecliptic (GSE) coordinates.

Data intervals of 12 h duration were taken from the following types of solar wind, with the number of events for each solar wind type given in parenthesis: 1) slow wind (55), 2) fast wind (49), 3) SIR (70), 5) ICME-driven sheath regions (27), and 7) magnetic clouds (74). The lower number of sheath events is explained by the requirement to have the interval duration at least 12 h. We note that sheaths and SIRs may in particular have some significant variations in their properties over the interval (e.g., Kilpua et al., 2017a), but we did not separate them into sub-intervals in order to have as long as possible durations for investigation. In SIRs, the stream interface (SI) separates the cooler, denser and slower solar wind from the tenuous, hot and fast wind (e.g., Gosling et al., 1978; Richardson, 2018). In sheaths, the field and plasma close to the shock has been recently processed by the shock, while close to the ICME leading edge plasma and field have evolved considerably since encountering the shock and could have been further modified by processes at the sheath-ICME boundary, e.g. field line draping. Both sheaths and SIRs are compressive structures, but a typical SIR at 1 au is not preceded by a shock (Jian et al., 2006). We here focus on the subset of ICMEs called magnetic clouds, which represent large-scale flux ropes ejected from the Sun (e.g., Burlaga et al., 1981; Klein and Burlaga, 1982).

The magnetic clouds were collected from the *Wind* ICME catalogue (Nieves-Chinchilla et al., 2018) and the Richardson and Cane ICME list (Richardson and Cane, 2010), the SIRs times from the ACE/WIND Stream Interaction Regions catalog, and sheaths from the list published by Kilpua et al. (2021). We here considered SIRs where the solar wind speed reached at least 650 km s$^{-1}$ after the SIR. Fast wind intervals were defined as the periods when the average wind speed during a 12 h interval after the SIR was $> 600$ km s$^{-1}$. The slow solar wind intervals were defined as the periods preceding the SIR during which the 12 h averaged wind speed was $< 450$ km s$^{-1}$.

### 2.2 Examples

Examples of solar wind data during each solar wind type are presented in Figure 1. The data series are shown only during the analyzed intervals and boundaries such as the shock and the ICME leading edge for the sheath are thus not included in the plots.





It can be seen that the magnetic cloud data series differs considerably from the other four solar wind types displayed. It exhibits smooth variations of all the three field components and a steady magnetic field magnitude. SIRs and sheaths exhibit the most variations in their properties and the field magnitude. The fast wind interval is characterized by the uniform presence of large-amplitude fluctuations that are typically taken to represent anti-sunward propagating Alfvén waves. The three bottom panels

of Figure 1 show white noise, pink noise and Brownian motion. These data series were created using the same sample length as the solar wind series above, i.e. 14,400 samples. The white and pink noise were created using a publicly available Python codes. White noise is a maximally random series where different frequencies have equal intensities. Its power spectral density is constant and gives a spectral slope of zero, i.e. for a $f^{-\alpha}$ power law, $\alpha = 0$. Pink noise is associated with the $f^{-1}$ spectrum; compared to white noise, it has relatively more power at low than high frequencies. In the solar wind, the $f^{-1}$ spectral range is

often interpreted as the 'energy containing' range, found at large scales where energy is believed to be injected in the system before cascading down the turbulent inertial range (e.g., Bruno and Carbone, 2013).

In the bottom panel, Brownian motion is shown for three values of the Hurst exponent. The Hurst exponent, $H$, is used to characterize the memory and correlations in the time series (e.g., Ruzmaikin et al., 1994). The exponent $H = 0.5$ describes the Brownian random walk (also called brown noise or classical Brownian motion) where the mean-squared distance from the

starting point of the walk increases as the square root of time. Such a time series is uncorrelated in the sense that the steps in the random walk are independent of each other or, in other words, that the auto-correlation of the time series is zero.

When $H \neq 0.5$ the process is called fractional Brownian motion (fBm). In such cases the increments in the random walk are not independent. The light brown curve in Figure 1 shows the case with $H = 0.8$. When $H > 0.5$ the time series is said to be persistent and exhibit long-term memory or long auto-correlation. The distance from the starting point in the random walk

increases faster than in the case of classical Brownian motion. The increasing (decreasing) value is followed by an increase (decrease) and the entropy is lower. The dark brown curve shows the case where $H = 0.2$. A time series with Hurst exponent $< 0.5$ is said to be mean-reverting or anti-persistent, i.e. an increase (decrease) is followed by a decrease (increase). Such short-memory series is unpredictable and has higher entropy and lower complexity when compared to brown noise or the case with $H > 0.5$. Now the distance from the starting point increases slower than for classical Brownian motion. A visual inspection of

the solar wind time series reveals that the magnetic cloud time series is most consistent with the larger Hurst exponents while the other intervals appear more like the short-memory fBm and pink noise.

### 2.3   Permutation entropy and Jensen-Shannon complexity

As discussed in Section 1, there are $d!$ possible permutations for the embedded dimension $d$. For example, if $d = 5$, there are 120 possible permutations: 12345 (indicating that samples in the series have a monotonously ascending order from beginning

to end), 13245, 12425, etc. If we denote the probability of permutation $j$ to be $p_j$, and the set of probabilities $P$, the permutation entropy according to Bandt and Pompe (2002) is defined as

$$S(P) = -\sum_{j=1}^{d!} p_j \log p_j$$





**Figure 1.** Example intervals. The top five panels show the interplanetary magnetic field components in GSE coordinates (blue: $B_X$, cyan: $B_Y$, purple: $B_Z$) and the field magnitude (black). The data shown is 3 s data from the *Wind* spacecraft. The three bottom panels show white noise, pink noise and Brownian motion for the same sample size (14,400 samples) as the solar wind intervals. Brownian motion is shown for three different values of the Hurst exponent, H= 0.2 (persistent), H= 0.5 (classical Brownian motion) and H= 0.8 (trend-reverting).



From above we see that $S(P) = 0$ when only one permutation occurs and it maximizes when all permutations occur with equal probability. The normalized permutation entropy

$H(P) = S(P)/\log d!$

is defined such that it takes values between 0 and 1, where 0 is the lowest entropy and 1 the maximum entropy.

The Jensen-Shannon complexity as defined by Rosso et al. (2007) is

$$C_J^S = -2\frac{S(\frac{P+P_e}{2}) - \frac{1}{2}S(P) - \frac{1}{2}S(P_e)}{\frac{d!+1}{d!}\log(d!+1) - 2\log(2d!) + \log d!}H(P)$$

The quantity $S(\frac{P+P_e}{2}) - \frac{1}{2}S(P) - \frac{1}{2}S(P_e)$ is the Jensen-Shannon divergence, which is a measure of similarity between two

probability distributions. In this case, $P$ is the distribution formed by the patterns in the investigated time series and $P_e$ is the distribution that maximizes the permutation entropy, i.e., the one where all permutations occur with equal probability. As stated in Section 1, both the perfectly random ($P = P_e$) and perfectly ordered ($H(P) \approx 0$) case yields zero complexity. In general, complexity is lower the closer the distribution $P$ is to the maximum entropy state and the lower the normalized permutation entropy $H(P)$. The highest complexity values require the repeated occurrence of certain patterns that reflect underlying chaotic

processes, i.e. they occur when distribution $P$ is far from $P_e$, but has higher normalized entropy.

The statistical robustness has been tested for the complexity-entropy analysis. The analyzed time series must be long enough so that enough permutation sequences can be extracted to allow all possible permutations to be sufficiently sampled. The commonly used robustness criteria are $N/d! > 5$ and $\sqrt{d!/(N - (d-1)r)} < 0.2$ (e.g., Osmane et al., 2019; Weygand and Kivelson, 2019) where $N$ is the number of samples in the investigated time series and $r$ gives the subsampling rate, i.e. the

time lag $\tau = r\Delta t$, where $\Delta t$ is the data resolution. Here the length of each time series is 12 h at 3 s resolution, thus giving 14,000 samples in each interval. We here use $d = 5$, similar to many previous studies of the solar wind (Weck et al., 2015; Weygand and Kivelson, 2019; Good et al., 2020). The embedded dimension of 5 gives $N/d! = 120$ and we limit our analysis to $r = 600$ ($\tau = 1800$ s). This gives $\sqrt{d!/(N - (d-1)r)} = 0.10$, indicating that both robustness criteria are well met.

## 3   Result

### 3.1   Ordinal patterns

We first investigate the occurrence of ordinal patterns in time series of the three GSE magnetic field components sampled across all events within the different solar wind categories. Figure 2 shows the distributions of median occurrence of permutations for three time lags $\tau = 180$, 900 and 1800 s (corresponding to sub-sampling rates $r = 60$, 300, and 900) and $d = 5$, i.e., each ordinal pattern has five elements. Note that a fixed number has been added to all curves except 'Magnetic Cloud' to aid comparison

between different categories (see the figure caption for details). The shaded areas show the interquartile ranges. For smallest timescales shown ($\tau = 180$ s), both the lower and upper quartiles are very close to the median while for the largest $\tau = 1800$ s values are more spread.





A clear trend visible in Figure 2 is that the permutations with several monotonically increasing and decreasing numbers (12345, 54321, 12354, 21345, etc.) are most abundant, in particular for $\tau = 180$ s for all investigated solar wind categories and magnetic field components. As the timescale increases such peaks become weaker for the fast wind in particular. For the magnetic clouds and also for sheaths the peaks become even more pronounced. Another striking feature in Figure 2 is that for $\tau = 180$ and 900 s in particular, the peaks and dips in the occurrence of permutations are highly correlated across all investigated solar wind categories and magnetic field components. The magnetic cloud category stands out as the one having certain permutations dominating also at the largest scales.

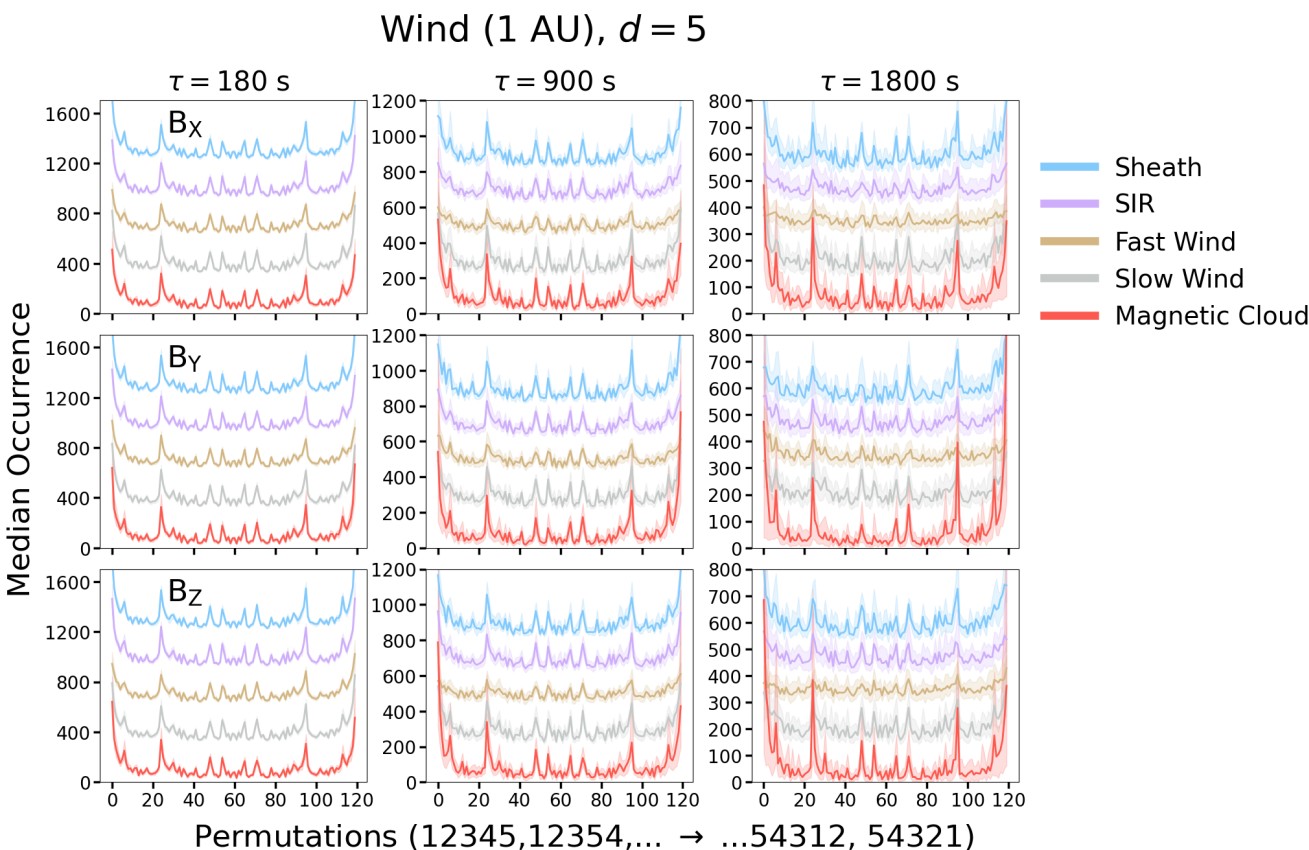

**Figure 2.** Median occurrence of permutations observed in different solar wind categories for $d = 5$ for $\tau = 180$, 900 and 1800 s, i.e. for subsampling rates $r = 60$, 300 and 600) for the three GSE magnetic field components. Note that all curves except the magnetic cloud curves have been shifted upwards by a fixed amount by 300 for $\tau = 180$ s, 200 for $\tau = 900$ s and 130 for $\tau = 1800$ s) to aid comparison. Shaded areas show the interquartile range.





### 3.2 Entropy and complexity of fluctuations

The (normalized) permutation entropy and complexity as a function of time lag $\tau$ (i.e. $r\Delta t$) are investigated in the two top panels of Figure 3 for $d = 5$. The time $\tau$ is increased from 60 s ($r = 20$) to 1800 s ($r = 600$) in steps of 60 s ($r = 20$). The uncertainty ranges indicated in the complexity panels are estimated from the average permutation occupation number $\sqrt{d!/(N-(d-1)r)}$ (e.g., Weygand and Kivelson, 2019; Good et al., 2020). The uncertainties increase with increasing time lag (i.e. with increasing $r$) as the total number of permutations extracted from the time series decreases slightly with decreasing $\tau$ ($r$).

The top panels of Figure 3 show that the entropies in the fast wind, slow wind, sheath and SIR categories show no or very weak dependencies on $\tau$, with entropy weakly reducing at larger $\tau$ in some components for the latter three categories. The magnetic cloud entropy, in contrast, shows a significant $\tau$ dependence, falling strongly in all three components at large $\tau$. For the fast wind the entropy in turn consistently increases very weakly with $\tau$ before flattening. These trends are similar for all magnetic field components. The key difference is that for magnetic clouds $B_Y$ and $B_Z$ components decrease to considerably lower entropies than $B_X$ at largest $\tau$. The values of entropy also differ between different solar wind categories at $\tau \gtrsim 300$ s. The fast wind consistently has the highest entropy and magnetic clouds the lowest entropy across all three components, with the other categories at intermediate values.

The complexity (second-row panels) broadly mirrors the entropy trends: with increasing $\tau$, complexity is approximately invariant in the fast wind, increases weakly in SIRs, sheaths and slow wind, and increases significantly in magnetic clouds. The relatively low entropy and high complexity in the magnetic clouds at large $\tau$ reflects their coherent, ordered structure at large scales, while the high entropy / low complexity of the fast wind reflects its unstructured, stochastic nature at all of the scales we have considered here. Again, the key difference between the components is for magnetic clouds. For the $B_Y$ and $B_Z$ components the complexity increases to larger values than for $B_X$ at largest $\tau$.

The third row of Figure 3 shows scatter plots of the average entropy and complexity values for the different solar wind categories. The time lags are shown from $\tau = 60$ s (lighter and smaller circles) to 1800 s (darker and larger circles). The bottom row of Figure 3 magnifies the high-entropy, low-complexity corner of the plots. The curves are shown for fBm with the Hurst exponent running from 0.05 to 0.75 in steps of 0.05, and for both time lags $\tau = 60$ s and $\tau = 1800$ s (sub-sampling rates $r = 20$ and $r = 600$). These plots show that the averaged values from the solar wind time series follow closely the fBm curves. In general, the averaged data points move towards the higher Hurst exponent ends of the fBm curves with increasing $\tau$ (increasing sub-sampling rates). The clearest exception is the fast wind. For the fast wind data points are clustered at the bottom-right corner and its data points exhibit higher entropies and lower complexities at the smaller smaller $\tau$. We also note that for most cases the data points for larger time lags are a bit above the fBm curve indicating a higher complexity.

### 3.3 Complexity–entropy map

In this section we investigate how the solar wind data series are placed onto the complexity–entropy plane, where the vertical axis shows complexity and the horizontal axis entropy (see e.g. Weck et al., 2015; Weygand and Kivelson, 2019). Highly





**Figure 3.** Entropy (first-row panels) and complexity (second-row panels) as a function of $\tau$ for $d = 5$, for the GSE magnetic field components (panels left to right) in the different solar wind categories (colour-coded lines). Shaded areas in the bottom panels give the uncertainty ranges estimated using the permutation occupation number approach. The third and fourth-row panels gives scatter plots of average entropy and complexity for the different solar wind categories from $\tau = 60$ to 1800 s in steps of 20 s. Time lag increases with the size and darkness of the marker circles. Two curves with black diamonds and squares show fractional Brownian motion for sub-sampling rates $r = 20$ and $r = 600$, respectively. The grey square and diamond markers show the curves at Hurst exponent 0.5.





stochastic fluctuations are represented by white and pink noise, which appears in the very bottom-right part of the plane with entropy $\sim 1$ and complexity $\sim 0$. Chaotic fluctuations have entropies between $\sim 0.45 - 0.70$ and complexities close to the

maximum complexity curve (e.g., Zanin and Olivares, 2021). Periodic fluctuations (e.g. sinusoidal functions) would fall onto the lower left part of the plane (not shown). They have low entropies $\sim 0 - 0.50$ and, while their complexities follow the maximum complexity curve, they do not attain the peak values. Given that differences between the GSE components were found to be relatively small in section 3.1, we here choose to investigate $B_Z$ only. $B_Z$ is also the IMF component that has most interest for geoeffectivity (e.g., Kilpua et al., 2017b).

Figure 4 shows the distribution of computed values for different solar wind categories in the complexity–entropy plane for time lags $\tau =, 180, 900$, and $1800$ s (subsampling rates $r =, 60, 300\ 900$). The dark blue-coloured dots correspond to $\tau = 180$ s, gray dots $\tau = 900$ s and the yellow dots $\tau = 1800$ s. The fBm points for varying Hurst exponents are also shown in the figure, for three investigated time lags / subsampling rates. The two black curves show the minimum and maximum complexity curves.

Magnetic clouds clearly exhibit the widest spread of data points in the complexity–entropy map. Their entropies reach to the

H$= 0.8$ markers, and complexities of about 0.35. While data points for $\tau = 180$ s fall onto the fBm curve, a significant fraction of data points deviates from the fBm curve for time lags $\tau = 900$, and $1800$ s. There is a clear trend that the data points move to lower entropies and higher complexities with the increasing time lag.

The fast wind data points are in turn clustered at the lower right part of the map, with the majority of them having entropies $\gtrsim 0.96$. For the slow wind and compressive structures (i.e. sheaths and SIRs), the lowest entropy values are $\sim 0.8$. For other

solar wind structures the organization with $\tau$ is less clear than for magnetic clouds. While the data points furthest along the fBm curves are solely related to the largest time lag $\tau = 1800$ s, in the right lower corner there is a mixture of data points from all included time lags. In particular for the fast wind and SIRs this region is occupied with data points related to $\tau = 900$ s and $\tau = 1800$ s.

### 3.4 Hurst exponent

Finally, we calculate the Hurst exponents for the investigated time series. As introduced in Section 2.2, the Hurst exponent, $H$, is used to characterize Brownian motion, with $H \sim 0.5$ signifying an uncorrelated random walk with short-memory (general Brownian motion or Brown noise), $H < 0.5$ mean-reverting and $H > 0.5$ a persistent series with long-term memory and structures.

For a time series $x(t)$ the Hurst exponent is related to the first-order structure function $S_1(\tau)$ as follows (e.g., De Michelis

et al., 2016; Gilmore et al., 2002; Giannattasio et al., 2022):

$$S_1(\tau) = \langle |x(t+\tau) - x(t)| \rangle \sim \tau^H.$$

The exponent $H$ can therefore be determined from linear regression fits to $log_{10}(\langle |x(t+\tau) - x(t)| \rangle)$ as a function of $log_{10}(\tau)$. In practice, this means calculating auto-correlations of $x$ with time lags $\tau$. The Hurst exponent relates to the spectral slope $\alpha$ of the fluctuation power spectrum $f^{-\alpha}$ as $\alpha = 2H + 1$ (Mandelbrot, 1977). This gives $H = 0.33$ for the Kolmogorov scaling with





**Figure 4.** Complexity–entropy maps showing Jensen-Shannon complexity in $B_Z$ plotted against normalized permutation entropy for $d = 5$ and sub-sampling rates ranging from $r = 20$ to $600$ in steps of $60$ ($\tau = 60$ to $1800$ s in steps of $60$ s). The dark blue points show values for $r = 20$, with point colour darkening to yellow with increasing $r$. In the top left panel the grey square and diamond markers show the fractional Brownian motion calculated with $r = 20$ and $r = 600$, respectively, for Hurst exponents from $0.1$ to $0.8$. The markers repeated in all panels with black edges indicate $H = 0.5$. Minimum and maximum complexity curves are shown in black.



spectral slope $\alpha = 5/3$ (Kolmogorov, 1941) and $H = 0.25$ for the Iroshnikov-Kraichnan scaling with spectral slope $\alpha - 3/2$
(Iroshnikov, 1964; Kraichnan, 1965).

As the scaling properties of the time series may not be constant we use a local Hurst exponent estimated by sliding a time window through the analysed time series, as was done for example by De Michelis et al. (2016). The authors note that the window length needs to be at least ten times larger than the maximum scale investigated. We here use a window width of 18,000 s (5 h) sliding in 30 min steps through each 12 h time series that allows to have the largest investigated time lag as
1800 s (30 min). The Hurst exponents are determined over 360 s (6 min) wide time lag intervals ($\Delta\tau$) in $\tau = 30$ s steps from 60 s to 1740 s. These 360 s intervals are stepped forward in 90 s steps. The events where the standard error in the fitting was $> 0.05$ were removed from the analysis. We however note that there are now drastic differences in the results when all exponents are included (data not shown), suggesting that removal of the points does not have strong influence.

The results are presented in Figures 5 and 6 for the $B_z$ component. The values are averaged across all events. The top
panels of Figure 5 show the colour maps of the Hurst exponent values as a function of the mid point of the sliding-window $\Delta\tau$-range and the time from the start of the 12 h interval in bins as described above. The bins where the Hurst exponent is close to Kolmogorov (H $= 0.32 - 0.34$), Iroshnikov-Kraichnan (H $= 0.24 - 0.26$) and random walk values (H $= 0.49 - 0.51$) are outlined and hatched. The bottom panels of Figure 5 show the percentage of the events when the standard error related to the fitting was $> 0.05$, i.e. the percentage of the removed points. At smallest scales the fitting was good for all events, while at
the largest scales up to $\sim 10$ % had the standard error exceeding our threshold.

The different solar wind types have some clear differences in their Hurst exponents. The fast solar wind has clearly the lowest Hurst exponents matching the Kolmogorov scaling ($H \sim 0.33$) at the smallest timescales and then decreasing towards Iroshnikov-Kraichnan scaling ($H \sim 0.25$) with the increasing timescale range over which the exponents are calculated. For several bins for the fast streams and in the end part of the SIRs the Hurst exponents are $\lesssim 0.25$. This indicates shallow spectral
slopes that could partly stem from the inclusion of the part of the $f^{-1}$ range.

The largest average Hurst exponent values are found for the sheaths and in particular for magnetic clouds. For sheaths the the largest Hurst exponents are at the smallest scales and towards the end part of the sheath at mid-scales. For magnetic clouds the largest exponents are clustered at the largest $\Delta\tau$. The slow wind has Hurst exponents matching Kolmogorov's at mid-scales and the exponents then become steeper at smaller and larger scales. For sheath, SIR and fast wind the Hurst exponents decrease
with increasing timescale while for the magnetic clouds the trend is indeed the opposite. Figure 5 also reveals a strong locality in Hurst exponents (spectral slopes) for sheaths and SIRs. Sheaths have clearly largest Hurst exponents (steepest slopes) at their end parts, i.e. close to the leading edge of the driving CME ejecta than close to the shock. For SIRs the Hurst exponents in turn get shallower close to the end part of the SIR.

Probability density functions (PDFs) of the Hurst exponent for varying $\Delta\tau$ are shown in Figure 6. The PDFs here include
values combined over the whole 12 h intervals. Firstly, for all investigated cases the PDFs become considerably flatter, i.e., the values are spread over a much broader range with increasing timescale. For the fast wind a strong peak is present at the at the Hurst exponent representing the Kolmogorov scaling ($H = 0.33$) that then moves first to Kraichnan scaling ($H = 0.25$) and then to even lower values with the increasing timescale. Nearly all values are in the anti-persistent Hurst exponent regime





($H < 0.5$). The fast wind had largest percentage of cases with poor fit which could partly stem from the slopes reaching partly

the $f^{-1}$ range. The slow wind PDFs peak primarily at values between Kolmogorov and fBm lines for all timescales. The sheath and SIR distributions overall quite similar to the slow wind PDFs. The key differences are the bias towards the larger Hurst exponents (steeper slopes) at the smallest scales and flatter distributions at the largest scales. As discussed above the local Hurst exponent maps revealed strong locality for sheaths and SIRs. The PDFs at larger timescales for magnetic clouds have strong tail at large Hurst exponent values extending to the persistent $H > 0.5$ regime.

## 265    4    Discussion

We first studied the occurrence of ordinal patterns, i.e. permutations in different solar wind types (fast and slow wind, magnetic clouds, SIRs and ICME-driven sheaths). The ordinal patters were constructed here from the set of five data points separated by a time lag $\tau$/sub-sampling rate $r$. We found that their occurrences were very similar in particular for timescale of 180 s and 900 s with certain permutations showing distinct peaks for all solar wind types. The consistent occurrence of certain permutations

suggests the presence of coherent structures or waves in the data.

At the smallest scale 180 s for all investigated solar wind types, we also found a clear dominance of ordinal patterns with a small change between the subsequent values, i.e. implying the absence of sudden and large changes of the magnetic field. Such peaks persisted for sheaths, slow wind and in particular for magnetic clouds also for the largest time lag included in the analysis (1800 s, i.e. 30 minutes), while for the fast wind peaks already got less dominant at the time lag 900 s. The reason is

that at the largest timescales magnetic clouds become increasingly sensitive to the coherent flux rope rotation. The persistence of peaks at large scales for sheath regions and slow wind suggests some coherent global structuring also for these solar wind types.

The entropy and complexity values between different solar wind types were found to be very similar at the smallest timescales/sub-sampling rates up to $\tau \sim 300$ s (subsampling rate $r \sim 100$). This suggests a uniformity in the physical pro-

cesses operating at smaller scales, independent of the large-scale structure of the solar wind. The entropy and complexity remained also relatively stable as a function of $\tau$ for all other solar wind types except for magnetic clouds. For magnetic clouds the entropy values strongly decreased and complexity values strongly increased with increasing $\tau$.

The entropy being approximately constant across the $\tau$ range with a slight increase towards the largest timescales is a signature of stochastic fluctuations (e.g., Osmane et al., 2019). This trend was identified here in particular for the fast wind

that also had throughout the investigated $\tau$ range the highest entropy and lowest complexity values. These finding imply that the fast wind is the most stochastic in nature from the investigated solar wind types. This could stem from the fact that the fast wind is permeated by Alfvénic fluctuations which are inherently stochastic in nature.

The above described entropy and complexity trends found for magnetic clouds are in turn in agreement with our understanding of magnetic clouds as ordered structures featuring low magnetic field variability and smooth rotation of the field direction

over an interval of one day (e.g., Klein and Burlaga, 1982; Kilpua et al., 2017a) and as discussed above, strong bias toward ordinal patterns where the values were steadily increasing or decreasing. The reason why in magnetic clouds $B_Y$ and $B_Z$




**Figure 5.** (Top) Colour maps showing the values of the local Hurst exponent as a function of time (given in minutes from the start of the solar wind time interval) and the mid-point of the time-lag interval used to calculate the Hurst exponent (see text for details). The values corresponding the Kolomogorov type scaling (K41), i.e. H= 0.33, Iroshnikov-Kraichnan (IK) type scaling H= 0.25 and uncorrelated random walk (H= 0.5) are shown as bordered and hatched bins. (Bottom) Same as in the top panels but for Jensen-Shannon complexity.



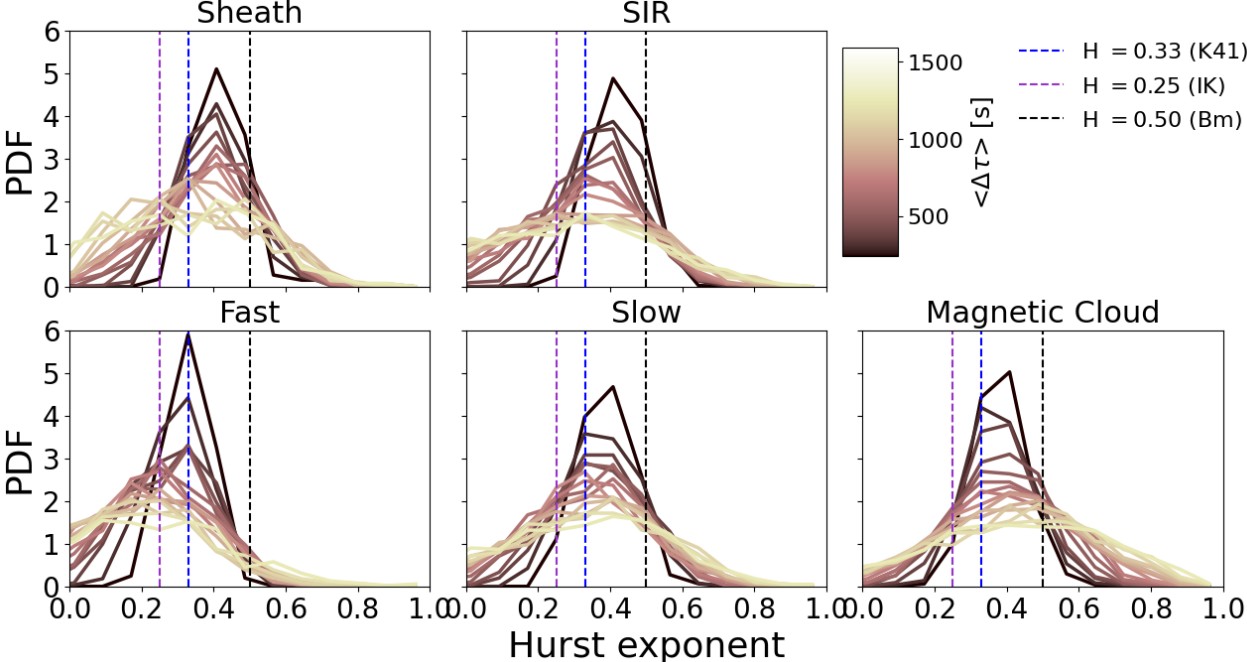

**Figure 6.** Probability density functions (PDFs) of the Hurst exponents calculated over varying time-lag ranges for different solar wind types.

components show higher complexities at larger timescales than $B_X$ is likely reflecting the fact that the large-scale and coherent magnetic field rotation in them occurs predominantly in $B_Y$ and $B_Z$. As mentioned earlier, their internal configuration is a magnetic flux rope. These structures propagate radially from the Sun, and therefore the minimum rotation of the field is in $B_X$.

295 Our finding that the fast wind had the higher entropy lower complexity than the slow wind is in contrast to Weygand and Kivelson (2019), but in agreement with Weck et al. (2015). The reason could be that Weygand and Kivelson (2019) used the *Ulysses* data at larger heliospheric distances while our study and Weck et al. (2015) use *Wind* data gather at 1 au. Fast wind is considered to be dynamically younger, and therefore expected to present less evolved turbulence (i.e., be less stochastic) than slow wind (e.g., Weygand and Kivelson, 2019). However, it could be that, at least near 1 au, slow wind has considerably

300 more coherent structures, while the fast wind is permeated by Alfvénic fluctuations which are inherently stochastic in nature. The intermittency of the fast wind is in fact known to increase with heliospheric distance from the Sun, while for the slow wind it remains approximately the same (e.g., Marsch and Liu, 1993). We also note that Weygand and Kivelson (2019) found complexities and entropies in interplanetary CMEs (ICMEs) to be similar to that of the slow wind, while in our study magnetic clouds had distinctly different values at larger time lags. We note that in addition including larger time lags, we separated

305 sheaths from the ejecta and include only magnetic clouds that are the most coherent subset of ICMEs. Weygand and Kivelson (2019) included ICMEs as a whole and they also included all ICMEs, not only magnetic clouds.

 The placement of data points in the complexity - entropy plane indicates that for most cases solar wind fluctuations follow relatively closely the fractional Brownian motion (fBm) curves. At the smallest scales the values fall exactly on the fBm curve,



while at larger scales there are some deviations, mostly above the fBm curve. This suggest higher ordering that likely stems
from the presence of coherent structures such as current sheets, small-scales flux ropes, and magnetic holes. These findings are
consistent with previous studies finding solar wind fluctuations being stochastic in nature, i.e the absence of chaotic or periodic
fluctuations. (e.g., Weck et al., 2015; Weygand and Kivelson, 2019; Good et al., 2020; Kilpua et al., 2022). It is however
possible that if larger timescales would have been included, the most coherent magnetic clouds would fall into the periodic
domain of the complexity - entropy plane as the time series of their magnetic field components resemble that of a half wave.

The fast wind exhibits least spread in their data points in the complexity - entropy plane and they are clustered closest to the
lower right part of the map close to the region where pink or white noise is located. This is in agreement with the previously
discussed results of the entropy and complexity analysis suggesting that the fast wind is highly stochastic in nature.

The widest spread of values in the complexity - entropy plane was observed for magnetic clouds for the largest time lags.
This could partly stem for a large variety of magnetic cloud structures observed in interplanetary space, from those exhibiting
distinctly smooth rotation to cases where considerable distortion is present (e.g., Kilpua et al., 2017a). However, for magnetic
clouds also, the data points at 180 s were clustered close onto the fBm curve, consistent with our previous suggestion of
uniformity of the processes at smaller scales in all solar wind types. The spread at larger time scales implies that magnetic
clouds are the least stochastic from the investigated time series and interpreting their nature in terms of the Hurst exponent
could therefore be questionable.

The relative similarity in fluctuation properties and placement in the complexity - entropy plane for the slow wind and the
large-scale compressive structures (sheaths and SIRs) could stem from the latter mostly consisting of the processed slow wind.
Although according to previous studies fluctuation properties change from the upstream to downstream at ICME-driven shocks
and some new fluctuations are created, they do not appear to reset the turbulence in a similar manner to planetary bow shocks
(Kilpua et al., 2021).

Both the entropy-complexity maps analysis and PDFs of Hurst exponents derived from the structure function analysis
showed a wide spread of Hurst exponents for the investigated time series at the larger time lags. It is also interesting to note
that for all other investigated solar wind types except for magnetic clouds most data points reaching the lowest right corner
of the complexity - entropy map, i.e. data points associated with the lowest Hurst exponents, were related to $\tau = 900$ s and
1800 s. In turn, all data points that were at the highest Hurst exponent values along the fBm curve were related almost solely
to the largest time lag 1800 s.

As discussed in Section 3.3 the Hurst exponent expresses whether fluctuations are persistent, random or anti-persistent.
Both the complexity - entropy maps and the Hurst exponents obtained from the structure function analysis indicate dominantly
anti-persistent fluctuations (H < 0.5) for all investigated solar wind types and timescales.

The exponents extending to the persistent regime (H > 0.5) were identified mostly in magnetic clouds and for the largest
time-scales. As mentioned previously, interpretation of Hurst exponents for magnetic clouds should be approached with cau-
tion. Visually (Figure 1) time series of magnetic field components extracted during magnetic clouds resemble persistent time
series with large Hurst exponent. We note that a significant fraction of magnetic clouds had however Hurst exponents close to
$H = 0.5$, i.e. being indicative of uncorrelated random walk. Using the relation $\alpha = 2H + 1$ this correspond to the inertial range



spectral slope $-2$, i.e. considerably steeper than Kolmogorov's. Recently, Good et al. (2023) showed that magnetic clouds in
the inner heliosphere can indeed exhibit slopes as steep as -2 at large scales. Rather than a property of the fluctuations, Good
et al. (2023) interpret these steep slopes as a feature of the background magnetic structure, with rotation of the global flux rope
field in the clouds adding power to the spectra at large scales.

For fast and slow wind the spectral slopes corresponding to the Hurst exponents are in agreement with previous studies
(e.g., Teodorescu et al., 2015; Borovsky et al., 2019; Yordanova et al., 2009). For example, we found a relatively wide range
of spectral indices, and shallower slopes for the fast wind than for the slow wind. The decrease in the Hurst exponents (slopes
getting increasingly shallower) with increase time lag for the fast wind is likely resulting from the inclusion of the $f^{-1}$ range.
The low-frequency break point for the fast wind occurs at relatively high frequencies (at 1 au frequencies corresponding a few
hours) while in the slow solar wind, it is found only when long enough (several days) time series are used (e.g., Bruno and
Carbone, 2013; Chen et al., 2020). The reason for steeper slopes for the slow wind could be related to increased intermittency
when compared to the fast wind. It should be noted that the conversion between the Hurst exponent and spectral slope $\alpha =$
$2H + 1$ assumes negligible intermittency (e.g., Giannattasio et al., 2022). In other words, the structure function analysis results
in steeper slope values than expected when intermittency is present. For the slow wind and compressive structures (SIRs and
sheaths) that are clearly stochastic in nature, steeper slopes reaching also $H = 0.5$ and even beyond for some events could be
related to intermittency steepening the slopes.

We found significant locality for the Hurst exponents for compressive SIRs and sheath structures. For sheaths the trailing
part had steeper slopes than the front part, while the for SIRs the trend is vice versa. Similar findings for sheaths were also
previously reported by (Kilpua et al., 2021) who compared spectral slopes between the leading and trailing parts of the sheath.
The larger Hurst exponents (steeper slower) could stem from highly fluctuating and more compressive fields present at the back
of the sheath near the flux rope's leading edge. The draping of the field lines around the flux rope, reconnection and depletion
regions can lead to current sheaths and discontinuities that can steepen the slope, i.e. increase close to $-2$. For magnetic clouds
the larger Hurst exponents at mid part of the cloud could stem from this region representing the least disturbed part of the
structure. The boundaries of magnetic clouds are often distorted by their interaction with the ambient solar wind (e.g., Kilpua
et al., 2013).

## 5   Conclusions

In this work, we have characterized time series sampled in fast and slow wind, magnetic clouds, CME-driven sheaths and SIRs.
The time series were analyzed by estimating their permutation entropy, Jensen-Shannon complexity and Hurst exponent from
the first-order structure function. The results reflect different dynamical processes behind the generation and evolution of the
solar wind structures and different behaviours with varying time scale. At small scales, all of the solar wind types show similar
occurrence frequency of ordinal patterns, entropy and complexity values, while clear differences are evident at large scales.
All solar wind types except magnetic clouds at largest scales follow relatively closely fractional Brownian motion (fBm) in
the complexity-entropy plane but are partly located at different parts of the time-scale dependent fBm curve. The fast wind

and magnetic clouds stood out as having the most distinct fluctuation characteristics, while the slow wind and compressive structures (SIRs and sheaths) resembled more closely each other. We also found a significant non-locality in Hurst exponents, in particular for sheaths and SIRs.

This study demonstrates that permutation entropy and complexity analysis is a useful tool for investigating the solar wind and its large-scale structures. The analysis can help to explore their internal processes, and how these internal processes relate to the local fluctuation properties. In addition, the complexity-entropy analysis could reveal the occurrence of mesoscale structures in space plasmas at different scales. Observations from the fleet of recent launched spacecraft (Solar Orbiter, Parker Solar Probe and BepiColombo) are also expected to yield important information on variations with heliospheric distance.

*Data availability.* The solar wind data used in this study are available from the NASA Goddard Space Flight Center Coordinated Data Analysis Web (CDAWeb; http://cdaweb.gsfc.nasa.gov/). The Wind ICME list is available at https://wind.nasa.gov/ICME_catalog/, the Richardson and ICME list at https://izw1.caltech.edu/ACE/ASC/DATA/level3/icmetable2.htm) (Richardson and Cane, 2010), and the ACE/WIND SIRs catalog at http://www.srl.caltech.edu/ACE/ASC/DATA/level3/SIR_List_1995_2009_Jian.pdf. The pink and white noise were generated using the code publicly available at https://github.com/felixpatzelt/colorednoise and the Fractional Brownian Motion with the package publicly 390 available at https://pypi.org/project/fbm/.

*Author contributions.* EK performed the data analysis and prepared the figures. All authors have contributed to the writing of the manuscript and interpretation of the results.

*Competing interests.* No competing interests are present.

*Acknowledgements.* We acknowledge the Finnish Centre of Excellence in Research of Sustainable Space (Academy of Finland grant number 395 312390). E.K. acknowledges the ERC under the European Union's Horizon 2020 Research and Innovation Programme Project SolMAG 724391. S.G. is supported by Academy of Finland grants 338486 and 346612 (INERTUM). M.A.-L. acknowledges the Emil Aaltonen Foundation for financial support.



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
