# Peer review of "Permutation Entropy and Complexity Analysis of Large-scale Solar Wind Structures and Streams"

_EGUsphere, 2023_

## Author Comment (AC1)

**General comment:**
The paper "Permutation Entropy and Complexity Analysis of Large-scale Solar Wind Structures and Streams" presents an extended characterization of the complexity of different solar wind structures such as ICMEs and SIRs along with fast and slow streams. Their stochastic character is investigated through permutation entropy and their complexity by means of the Jensen-Shannon complexity. The main results of this research is that plasma coherent structures such as ICMEs are the most complex (lowest entropy and highest complexity), whereas fast wind is the most stochastic (lowest complexity and highest entropy). Finally authors provide a local study of the Hurst exponent as a function of time and scales for the different stream categories. The paper is well organized, clear and very interesting. However at this stage I cannot recommend the publication as there are few issues that the authors should clarify.

**We thank the referee for their careful reading of our manuscript and constructive comments and suggestions. We have modified the paper, accordingly, please find our responses from below. The changed parts are shown bolded in the paper.**

**Specific comments:**

At paragraph 35, and throughout the paper, Authors present an extensive list of citations of previous permutation entropy applications in space physics studies. However, the paper by Raath et al. 2022 (https://doi.org/10.1029/2021JA030200open_in_new; R22) does not appear in the list. I recommend to include R22 in the bibliography, but, most importantly, to comment on the results, since R22 has several things in common with this study. For instance, in Fig. 12 of R22, Authors show a comparison between low-H data interval, fBM and ICME in the CH-plane. ICMEs are estimated through the method by Wang and Richardson 2004, thus compatible with the magnetic cloud set of this study. The analysis of R22 is based on Voyager 2 data and results are fairly in agreement with those of this study.

**Many thanks for pointing out this study that had slipped our attention. It is definitely relevant to this work, and we have now added Raath et al. 2022 as a reference and included discussion of the results both in the Introduction and in the Discussion.**

In the discussion Authors comment about the variability of the local Hurst exponent in terms of spectral slopes and intermittency. In my opinion there is something controversial in this part, since the relation between the Hurst exponent and the spectral slope $\beta=2H+1$ only holds under the condition of global self-similarity, as, for instance, for the fBM. Since the hallmark of turbulence is its multifractal character, conclusions about intermittency based on the Hurst exponent only cannot be fully consistent. The anomalous scaling, indeed, appears in high order structure functions $S_q$ (say $q > 4$) and, although it is well established that intermittency varies with heliospheric distance and also varies among streams of different nature, a discussion about turbulence/intermittency without inspecting measures based on high-order statistics (i.e., kurtosis) is incomplete. For example, Authors observe intervals in the time-scale diagram where the Hurst exponent matches the scaling predictions by Kolmogorov or Kraichnan. However, it is not possible to characterize the turbulence by only looking at the first-order scaling exponent also in these cases. Such intervals, indeed, are also strongly intermittent and therefore high-order statistical measures are needed. So, I recommend revising the discussions about Hurst exponent and turbulence and the

association between H and β, since the first scaling exponent $\zeta_1$ coincides with H if and only if the scaling law is linear, i.e., $\zeta_q \sim Hq$ (e.g., Flandrin, 1989).

**We have considerably modified this part in our paper. The paper indeed had too much focus on discussing the slopes considering that solar wind time series are expected to have multifractality. We now discuss this in more detail when introducing the Hurst exponent (now in Section 2.3). The presentation of the results and discussion now focuses on the Hurst exponents and persistent/anti-persistent aspects, not about the spectral slopes. When we discuss the slopes we now remind the reader that the relationship between the Hurst exponent and slopes should be taken with caution. Note that from Figure 5 we have also removed the hatched region indicating the scaling exponents not to give too much emphasis on this relation.**

**Minor corrections:**
Abstract: Since part of the discussion of the paper is made on the local Hurst exponent, I would recommend to mention this in the abstract

**We have added now mentioning of the Hurst exponent in the abstract**

Line 104-105: Regarding the Brownian motion, please correct "square-root of time" with "time".

**corrected**

Hurst exponent and permutation entropy are both indicated with H throughout the paper. I would recommend using different symbols for them to avoid any confusion.

**Text indeed needs to be modified to have different symbols for the permutation entropy and Hurst exponent. We have changed the permutation entropy symbol as calligraphy H_S to stand out from H signifying the Hurst exponent.**

Line 115: Authors write that "magnetic cloud time series are more consistent with larger Hurst exponents", but also slow wind time series appear more similar to magnetic cloud than sheats or fast wind intervals.

**Yes, that seems a fair judgment. We have modified the text accordingly.**

Line 128: There is an extra factor H(P) in the definition of the Jensen-Shannon complexity measure, please remove it.

**This equation is equal to the form given e.g. in Rosso et al., 2007 and Osmane et al. 2019 where the Jensen-Shannon complexity is defined as C_JS = D x Hs(P), where D is the Jensen-Shannon divergence and Hs (P) the normalized Shannon entropy.**

Line 147 and 201: 900 –> 600

**corrected**

Line 219: Authors write "the Hurst exponent is related to the first-order structure function as follows". I would recommend to state explicitly that this relation holds if the scaling-law is linear, since «H is determined practically, by plotting $S_q(\tau)$ vs $\tau$ on a log-log plot, and taking the slope, which is equal to qH», as stated in Gilmoure et al. 2002.

**Modified accordingly.**

Line 246: delete the extra-"the" at the end of the row

**corrected**

Line 324: Authors write that in the case of magnetic cloud "interpreting their nature in terms of the Hurst exponent could therefore be questionable". Please explain why.
**The Hurst exponent analysis should assume that time-series are stochastic. But indeed, even though least stochastic, magnetic clouds a still according to CH-analysis stochastic. This has now been clarified and written less strongly.**
Line 382: Authors conclude that "complexity-entropy analysis could reveal the occurrence of mesoscale structures in space plasmas at different scales". How this statistical analysis can be used to identify mesoscale structures? What the authors mean by mesoscale structures in this context?

**Mesoscale structures have a wide range definition in the literature from a few hundred of Earth radii up to about 0.01 au (the scales are given in the Introduction). The applicability depends on the length of the used time-series. We mean this in the statistical sense, where frequent occurrence of mesoscale structures could add more structures to the time-series that is detected as larger complexity. We have now clarified this in the text.**

**Citation**: https://doi.org/10.5194/egusphere-2023-2352-RC1

---

## Author Comment (AC2)

**We thank the reviewer for careful reading of our manuscript and constructive comments and suggestions. We have modified the paper, accordingly, please find our responses from below. The changed parts are shown modified in the paper.**

This work aims at characterizing time series of fast and slow solar wind, magnetic clouds, CME-driven sheaths and SIRs using permutation entropy, Jensen-Shannon complexity and Hurst exponent analyses. The study is original and innovative and worthy of prompt publication in ANGEO, following a few minor revisions that mainly concern adding clarifications and pertinent references in the manuscript, as well as reorganizing its structure.

1.    Introduction (in agreement with Referee #1 remark on Abstract): Since part of the discussion of the paper is made on the local Hurst exponent, I would recommend devoting a paragraph to discussing this in the Introduction. For instance, I feel it would be fair to mention one of the first studies that used the Hurst exponent to study the geospace and specifically the geomagnetic activity, which was published in the same journal as the present manuscript under review (Balasis et al., 2006). In Balasis et al. (2006), the transition from anti-persistent to persistent behavior was associated with the occurrence of intense magnetic storms. Moreover, entropy analysis has also been used in several publications to study the near-Earth electromagnetic environment (for a recent review see Balasis et al., 2023).

**We have added Balasis et al. 2006 and 2023 as a reference. We also now discuss the Hurst exponent in the Introduction where these references occur. We thank the reviewer for pointing these relevant papers out.**

2.    Subsection 2.1: what is the time interval covered by the data considered in this study? For instance, do you analyze time series covering a full solar cycle? Please make this point clear here.

**This was indeed missing from the paper. We have now added the years in Section 2.1 (1997 – 2022) and mention that this period covers two solar cycles.**

3.    Subsection 2.2: at this instance the Hurst exponent along with the fBm model suddenly jumps into the manuscript to characterize the various types of solar wind time series. I think it would be making more sense to introduce the Hurst exponent together with the theory of the other analysis techniques of Permutation entropy and Jensen-Shannon complexity (as given in 2.3) and then move to Figure 1 together with the Results section. It is rather awkward to first apply the Hurst exponent and then introduce the related theory in Subsection 3.4. So, in my opinion, 2.3 and 3.4 should be combined in a common methodological section and presented before Section 3 of the Results.

**This is a good suggestion to improve the logical structuring of the paper. We now mention Hurst exponent for the first time already in the Introduction and have moved its more detailed description from Section 3 into a new subsection in Section 2. Examples have now been moved to the beginning of Section 3.**

4.    Lines 284–285 read: "This trend was identified here in particular for the fast wind that also had throughout the investigated τ range the highest entropy and lowest complexity values." I am a bit confused, if I understood well, with the suggested link between the highest entropy and lowest complexity, since in my (traditional?) perspective higher entropy values mean a lower organization or a less ordered state of the system under study, which in turn points to higher complexity values also. Therefore, higher entropy means higher complexity! Could you please comment upon this point?

**Complexity according to its definition is high for purely ordered and random time-series that give zero complexity. This is explained in Section 2.3 of the paper. It is related to the Jensen-Shannon divergence, which is a measure of similarity between two probability distributions.**

5.    Lines 285–286 read: "This could stem from the fact that the fast wind is permeated by Alfvénic fluctuations which are inherently stochastic in nature." Why is that happening? please elaborate / explain a bit this point.

**Alfven waves are stochastic fluctuations in the solar wind and they are primarily observed in the fast wind. We have rewritten this part in the manuscript.**

6.       Last but not least, lines 339-340 read: "The exponents extending to the persistent regime (H > 0.5) were identified mostly in magnetic clouds and for the largest time-scales." In previous Hurst exponent studies of geomagnetic activity indices, as well as corresponding solar wind variations (e.g., Balasis et al., 2006), persistence was associated with the occurrence of intense magnetic storms, i.e., with an extreme event. What could be a possible extreme event in your case?

**This is a very interesting question! We  guess that in a solar wind context extreme events could be considered to the Sun generating big eruptions, such as magnetic clouds, instead of a more consistent background of in particular the fast wind. Our analysis however includes time-series recorded only within the structures so in the present study the results tell more about the generation of fluctuations or smaller scale sub-structures within these structures. Magnetic clouds having high Hurst exponents are related likely to values in them having tendency to increase or decrease in coherent manner. In slow wind and sheath time series, an increased Hurst exponent could reflect the smaller scale transiently generated structures. We have included a brief contemplation of this with the reference to Balais et al., 2003 at the end of the Discussion section.**